# Galectin-2 Has Bactericidal Effects against *Helicobacter pylori* in a β-galactoside-Dependent Manner

**DOI:** 10.3390/ijms21082697

**Published:** 2020-04-13

**Authors:** Takaharu Sasaki, Rei Saito, Midori Oyama, Tomoharu Takeuchi, Toru Tanaka, Hideshi Natsume, Mayumi Tamura, Yoichiro Arata, Tomomi Hatanaka

**Affiliations:** 1Faculty of Pharmacy and Pharmaceutical Sciences, Josai University, 1-1 Keyakidai, Saitama 350-0295, Japan; gyd1702@josai.ac.jp (T.S.); yy14133@josai.ac.jp (R.S.); oyamami@josai.ac.jp (M.O.); t-take@josai.ac.jp (T.T.); tanakato@josai.ac.jp (T.T.); natsume@josai.ac.jp (H.N.); 2Faculty of Pharma-Science, Teikyo University, 2–11–1 Kaga, Itabashi-ku, Tokyo 173–8605, Japan; m-tamura@pharm.teikyo-u.ac.jp (M.T.); arata@pharm.teikyo-u.ac.jp (Y.A.); 3Tokai University School of Medicine, 143 Shimokasuya, Isehara, Kanagawa 259–1193, Japan

**Keywords:** galectin-2, *Helicobacter pylori*, immunoreaction, bactericidal effect

## Abstract

*Helicobacter pylori* is associated with the onset of gastritis, peptic ulcers, and gastric cancer. Galectins are a family of β-galactoside-binding proteins involved in diverse biological phenomena. Galectin-2 (Gal-2), a member of the galectin family, is predominantly expressed in the gastrointestinal tract. Although some galectin family proteins are involved in immunoreaction, the role of Gal-2 against *H. pylori* infection remains unclear. In this study, the effects of Gal-2 on *H. pylori* morphology and survival were examined. Gal-2 induced *H. pylori* aggregation depending on β-galactoside and demonstrated a bactericidal effect. Immunohistochemical staining of the gastric tissue indicated that Gal-2 existed in the gastric mucus, as well as mucosa. These results suggested that Gal-2 plays a role in innate immunity against *H. pylori* infection in gastric mucus.

## 1. Introduction

*Helicobacter pylori* is a helix-shaped Gram-negative bacterium with high motility. It was first identified in the stomach of chronic gastritis patients by Marshall and Warren in 1983 [1]. Until the discovery of the bacterium, the stomach had been regarded as a sterile organ due to its acidic internal environment. *H. pylori* expresses urease, which hydrolyzes urea to ammonia and carbon dioxide, and this enzyme allows the bacterium to survive in the acidic environment of the stomach.

*H. pylori* can cause several diseases including gastritis, peptic ulcer, mucosa-associated lymphoid tissue lymphoma, and gastric cancer [2,3]. As the bacterium was categorized as ‘Group 1′ (which is carcinogenic to humans) by the International Agency for Research on Cancer [4], the major drug therapy for gastritis and peptic ulcer has turned to antibiotics against *H. pylori* from the use of antacids. A combined drug therapy consisting of clarithromycin, amoxicillin, and proton pump inhibitors is covered by Japan’s National Health Insurance to eradicate the bacterium in positive patients. Although the widespread use of this therapy reduced the morbidity of *H. pylori*-mediated peptic ulcers and gastric cancer [5,6], the bacterium eradication efficacy remained approximately 70% due to the appearance of clarithromycin resistant bacterium [7]. The bactericidal efficacy of secondary therapy substituting metronidazole for clarithromycin is approximately 80% [8] and the value is suspected to decrease in the future due to the speculated appearance of the antibiotic-resistant bacterium [9]. In addition, amoxicillin cannot be prescribed to patients with penicillin allergy. Therefore, the development of novel therapeutic strategies with a higher bactericidal efficacy and applicability to a wide range of *H. pylori*-positive patients is desired.

Galectins are a family of animal lectins with affinity to β-galactosides of glycoconjugates and are involved in a wide variety of biological phenomena such as cell adhesion, migration of leukocytes, differentiation, and apoptosis. Fifteen members of the galectin family proteins have been found in mammalian species [10]. Galectins are classified into three types based on their molecular architectures, i.e., proto-type, chimera-type, and tandem repeat-type; proto- and chimera-types have one carbohydrate recognition domain (CRD) for β-galactosides in a molecule and the tandem repeat-type has two CRDs [11]. Proto-type galectins can form noncovalent symmetric dimers depending on their concentrations and the surrounding environment. The other galectins are also capable of forming dimers or oligomers. Thus, galectins can crosslink glycoconjugates possessing β-galactoside structures.

*H. pylori*, similar to other typical Gram-negative bacteria, possess an outer membrane consisting of phospholipids and lipopolysaccharide (LPS), with fucosylated O-antigen of LPS [12]. The fucosylated O-antigen of *H. pylori* mimics Lewis antigens found on the epithelial cells and mucins in the human stomach, allowing *H. pylori* to avoid the host immunity [13,14]. These LPS can be crosslinked by galectins. Notably, some galectin family proteins such as galectin-3 (Gal-3), -4, and -9 are expressed in the gastrointestinal tract, recognize pathogens, and kill them [15]. Gal-3 is involved in innate immunity by inducing the aggregation of *H. pylori* and then killing the bacteria in an O-antigen-dependent manner [16,17].

Gal-2 is localized in gastrointestinal epithelium cells and is specifically expressed in surface mucous cells and mucous neck cells in the stomach [18,19,20]. Previously, our in vitro study suggested that Gal-2 could strengthen the barrier structure of the gastric mucosa by crosslinking mucins [21]. Acute and chronic colitis in mice was ameliorated by Gal-2 overexpression [22]. In human gastric cancer tissues, Gal-2 expression is decreased by lymph node metastasis of gastric carcinoma [23], with reduced Gal-2 expression observed at mice lesion sites bearing *Helicobacter*-induced gastric cancer [24]. Like Gal-3, Gal-2 may play several crucial roles for biophylaxis against *H. pylori* infection.

In this study, we assessed whether Gal-2 was involved in host immunity against *H. pylori*. The effect of Gal-2 on the aggregation and growth of *H. pylori* were observed and the β-galactoside-dependency was investigated. The bactericidal effect of Gal-2 on *H. pylori* was evaluated by using fluorescence dyes to specifically stain live or dead bacterial cells. Moreover, the distribution of Gal-2 in the gastric mucus was examined to identify the potential interacting site of Gal-2 with *H. pylori* in vivo.

## 2. Results

### 2.1. Gal-2 Induces Aggregation of H. pylori

To verify whether Gal-2 affected the formation of *H. pylori* aggregates, the bacterial suspension was mixed with Gal-2 for 1 h and then observed under an optical microscope (Figure 1). Several clumps of *H. pylori,* of different sizes and shapes, were formed following the addition of rat Gal-2 (rGal-2) as shown in Figure 1A; no bacterial aggregation was observed following the addition of phosphate-buffered saline (Figure 1B). The relationship between the *H. pylori* aggregation and Gal-2 concentration was investigated by counting the nonaggregated bacteria owing to the nonuniformity of clumps (Figure 1B). The number of nonaggregated bacteria was reduced with increasing Gal-2 concentrations. The effect of human Gal-2 (hGal-2) was approximately comparable to rGal-2, that is, no great species difference was observed between rats and humans in the *H. pylori* aggregation effects induced by Gal-2.

### 2.2. Gal-2-H. pylori Interaction Depends on β-galactosides

To evaluate whether the aggregation of *H. pylori* by Gal-2 was induced via the recognition of β-galactoside-containing glycoconjugates on the bacteria, the effect of competitive sugars on the Gal-2 dependent aggregation of *H. pylori* was investigated (Figure 2). The addition of 0.1 M lactose, which contains a β-galactoside structure, to the *H. pylori* suspension inhibited rGal-2 and hGal-2 dependent bacterial aggregation; sucrose, which has no β-galactoside structure, failed to inhibit aggregation. No marked species differences were observed between rats and humans in the β-galactoside dependency.

To confirm that Gal-2 exists in the *H. pylori* clumps, a green fluorescent protein (GFP)-tagged rGal-2 was added, instead of Gal-2, to the *H. pylori* suspension, which was then observed under optical and fluorescence microscopes (Figure 3). In the *H. pylori* suspension (control), no clumps and fluorescence were observed (Figure 3A). When rGal-2-GFP was added to the suspension, bacterial aggregation was noted and the fluorescence from rGal-2-GFP was located on the clumps (Figure 3B). The phenomena induced by the addition of rGal-2-GFP disappeared under lactose coexistence conditions (Figure 3C), whereas sucrose failed to influence this phenomenon (Figure 3D). These results suggest that Gal-2 induced the aggregation of *H. pylori* through the crosslinking of β-galactoside-containing glycoconjugates on the bacteria.

### 2.3. Gal-2 Inhibits H. pylori Growth

The influence of Gal-2 on the growth of *H. pylori* was evaluated by increased turbidity of bacterial suspension (measured as optical density ratio at 600 nm), before and after incubation with the protein for 3 days (Figure 4). Higher the Gal-2 concentration, the stronger the inhibitory effect on bacterial growth. Both rGal-2 and hGal-2 significantly inhibited bacterial growth at concentrations exceeding 93.9 µg/mL.

### 2.4. Gal-2 has Bactericidal Effects against H. pylori

To evaluate the bactericidal effect of Gal-2, the *H. pylori* suspension was incubated with Gal-2 for 1 h and then stained with carboxyfluorescein diacetate (CFDA) and propidium iodide (PI), for live and dead cells, respectively (Figure 5). In the control, only green fluorescence of CFDA was observed under the fluorescence microscope and thus most bacteria observed under optical microscopes were alive (Figure 5A). When *H. pylori* was incubated with Gal-2, the red fluorescence of PI was observed (Figure 5B,C). The central region of the clumps was stained with PI, although the cells surrounding the clumps were barely stained with CFDA. Conversely, most nonaggregated bacteria were stained with CFDA. These results suggested that rGal-2 and hGal-2 showed bactericidal effects against *H. pylori*.

### 2.5. Gal-2 Distributes in Gastric Mucus

Distribution of Gal-2 in the gastric mucosa and mucus of mice was evaluated by periodic acid-Schiff (PAS) staining and immunohistochemistry (IHC) with the anti-galectin-2 antibody. The mucosa and mucus of mice were simultaneously fixed using Carnoy’s solution. PAS staining visualized the mucus that existed just above the mucosa (Figure 6A). Gal-2 was observed in the mucus and mucosa by IHC (Figure 6C), which could not be considered a nonspecific reaction (Figure 6B). It is possible that Gal-2 induces the aggregation of *H. pylori* and then demonstrates the bactericidal effect in gastric mucus.

## 3. Discussion

The immune response against pathogens is one of the important biological functions mediated by galectins. The pathogen recognition and killing abilities of galectin-1, -3, and -8 have been extensively investigated; however, data on Gal-2 is lacking [15]. In this study, we observed that Gal-2 could be involved in host immunity against *H. pylori* infection. The aggregation of *H. pylori* was induced by Gal-2 through crosslinking of the β-galactoside structures on the bacteria, producing a bactericidal effect.

The aggregation of *H. pylori* depended on the concentration of Gal-2 (Figure 1). Gal-2 can crosslink β-galactoside structures by forming homodimers [25]. The crosslinking via β-galactoside-containing glycoconjugates on *H. pylori* can induce bacterial aggregation. The aggregation induced by Gal-2 was inhibited by lactose, which possesses a β-galactoside structure (Figure 2). When rGal-2-GFP was added to the *H. pylori* suspension, fluorescence was detected on the clumps (Figure 3). Gal-2 may induce *H. pylori* aggregation by crosslinking LPS on the bacterial cell surface. *H. pylori* (ATCC43504) has Lewis X, Lewis Y, and H type I within the LPS as O-antigens [12]. Reportedly, Gal-3 binds to the O-antigen of *H. pylori* [16], with Gal-2 demonstrating an affinity to H type I in vitro [26]. It is possible that Gal-2 crosslinks H type I expressed on *H. pylori* cells, producing clumps. Further investigations are imperative to elucidate the detailed mechanism.

Additionally, several free *H. pylori* cells, which were not involved in the aggregation induced by Gal-2, were observed (Figure 1). Following the addition of rGal-2-GFP, no fluorescence was detected on the free *H. pylori* (Figure 2). These *H. pylori* may possess minimal or no β-galactoside-containing glycoconjugates. Furthermore, even if rGal-2-GFP bound to these *H. pylori*, the fluorescence may be extremely weak to be detected by the fluorescence microscope owing to the low density of the protein. *H. pylori* has various types of carbohydrate chains containing the O-antigen. The O-antigen expression of *H. pylori* is altered by the environment surrounding the bacteria, e.g., pH and host immunity [27,28]. The expression fluctuates based on the LPS phase variation, even in an identical colony [29]. Thus, free bacteria may have little or no β-galactoside-containing glycoconjugates.

The aggregation of *H. pylori* by Gal-2 could occur as follows: Gal-2 binds to β-galactoside-containing glycoconjugates of a bacterium via one CRD of the dimer; the Gal-2-binding bacterium binds to another bacterium with or without Gal-2 via the another CRD of the dimer. No great species differences were observed in the aggregation effects induced by Gal-2 between rats and humans (Figure 1), consistent with the similarities observed in their Gal-2 amino acid sequence [30].

Gal-2 inhibited *H. pylori* growth and demonstrated a bactericidal effect (Figure 4; Figure 5). *H. pylori* in the center of clumps induced by Gal-2 were dead, with the bacterium in the surrounding barely survived. Gal-3 induces ATP metabolic disorders and morphological alterations in the *H. pylori* cell wall [17]. Gal-4 and Gal-8 kill blood group B positive *Escherichia coli* by disrupting the bacterial membrane morphology [31]. It can be postulated that Gal-2 disrupts the metabolic functions and membrane integrity of *H. pylori*. Assuming that Gal-2 has a bactericidal effect by binding to the β -galactoside structure of *H. pylori*, the more the protein binds to the bacteria, the greater the bacterial damage. Thus, the dead bacteria were observed in the center of clumps where the Gal-2 concentration was high. Another explanation for the bactericidal effect of Gal-2 is based on the aggregation of *H. pylori* itself. The survival of *H. pylori* requires the brain heart infusion broth, a highly nutritious general-purpose growth medium, and a microaerophilic environment. Hence, the center of the clumps might possess inadequate conditions for survival. To understand a detail molecular mechanism for the bactericidal effect of Gal-2, further studies including metabolome analysis are required.

The immunoreactivity of Gal-2 overlapped the PAS-positive region around the mouse mucosa; hence, Gal-2 existed in the mouse gastric mucus, as well as mucosa (Figure 6). The growth of *H. pylori* was inhibited by Gal-2 at concentrations above 93.9 µg/mL (Figure 4). Although the concentration of Gal-2 in the gastric mucosal epithelium is unknown, the protein exists at about 15 ng/mL in the serum [32]. In the spleen, the galectin-1 expression is 35–40 µg/mL, which is much higher than in the serum [32,33]. Furthermore, it is speculated that the Gal-2 expression in gastrointestinal cells is substantially higher than in the serum. Expression levels of galectins alter dramatically under various conditions, including pathogenic infections or disease progression [34]. Gal-3 expression in the stroma of the stomach is elevated following an *H. pylori* infection [35]. Therefore, we anticipated that the Gal-2 concentrations used in this study were within physiological limits. Because the aggregation effect of Gal-2 was confirmed within pH 5 to 7 (data not shown), the clumps of *H. pylori* can be formed in the mucus close to mucosa. There are a lot of glycoconjugate-containing substrates in the mucus and the substrate may affect the interaction of Gal-2 and *H. pylori*. Gal-3 was reported to produce *H. pylori* clumps on the mucosal surface [17]. Induction of aggregation, growth suppression and bactericidal effects of Gal-2 against *H. pylori* could occur in the gastric mucus in vivo. Further study is required to confirm these effects in vivo.

In conclusion, Gal-2 demonstrated the induction of aggregation, growth inhibition, and bactericidal effects in *H. pylori* and this protein exists in the gastric mucus. These results suggest that Gal-2 is involved in the immune response for the *H. pylori* infection. These findings could aid in the development of novel antibiotics and therapeutic strategies against *H. pylori* infection. Although the structural modification of Gal-2 using protein engineering or formulation design for suitable drug delivery might be necessary to prevent side effects such as ischemia risk for coronary artery disease patients [36,37], Gal-2 can be used as a substitute to conventional antibiotics.

## 4. Materials and Methods

### 4.1. H. pylori and Growth Condition

The *H. pylori* strain, ATCC43504, was purchased from the American Type Culture Collection (ATCC, Manassas, VA, USA) and the bacteria were stored at –80 °C as a glycerol stock solution. The dissolved stock was inoculated into a selective agar medium (Nissui Pharmaceutical, Tokyo, Japan) and incubated in a jar conditioned with Anaeropack microaerophilic (Mitsubishi Gas Chemical, Tokyo, Japan) at 37 °C for 4 days. The colonies that appeared were employed for subsequent experiments and subculture.

### 4.2. Preparation of H. pylori Suspension

*H. pylori* colonies were harvested and suspended in Hank’s balanced salt solution (HBSS, Invitrogen, Carlsbad, CA, USA). Nonsuspended bacteria were removed by centrifugation. The supernatant was adjusted to 1.0 of optical density at 600 nm (OD_600_). The suspension was used in the following experiments. Approximately, 10^8^ colony-forming unit (CFU)/mL corresponded to 0.5 of OD_600_.

### 4.3. Purification of Recombinant Galectin Proteins

The expression and purification of the recombinant rat Gal-2 (rGal-2), green fluorescent protein-tagged rat Gal-2 (rGal-2-GFP), and human Gal-2 (hGal-2) were performed using pET21a vector (Novagen, Merk KGaA, Darmstadt, Germany) as previously described [38,39]. Briefly, *E. coli* cells were incubated and subjected to protein expression by adding isopropyl-β-thiogalactopyranoside. These cells were harvested and sonicated; the debris was removed by centrifugation. The recombinant protein was purified by adding the extract to the β-galactoside-immobilized column. To remove LPS, Detoxi-Gel (Thermo Fisher Scientific, Fremont, CA, USA) was added to the purified protein solution and the mixture was incubated at 4 °C for 30 min with mixing. The resin was removed by centrifugation. The supernatant was filtered at 0.2 µm and used for experiments.

### 4.4. Aggregation Assay

The aggregation assay was performed as previously described with a slight modification [40]. Briefly, 100 µL of the *H. pylori* suspension was added to the equivalent Gal-2 solution and incubated for 1 h at 37 °C. After the mixture was vortexed for 2500 rpm, the sample (2 µL) obtained from the mixture was loaded on a microscope slide and covered with a coverslip (1.8 mm × 1.8 mm). The sample was observed using a microscope (BX-51, Olympus, Tokyo, Japan, 40x/numerical aperture; 0.95) equipped with a digital camera (DP-26, Olympus), and five images were randomly taken. The concentration of nonaggregated bacteria was calculated as follows: the number of bacteria/µL = the number of bacteria per constant area (0.2 mm × 0.2 mm) × 40.5. Nonaggregated bacteria were determined as described above [41]. In brief, bacteria existing alone or two linearly connected bacteria were classified as nonaggregated. Aggregation effects of Gal-2 were also investigated under 0.1 M lactose or sucrose conditions.

### 4.5. Inhibitory Activity of Gal-2 against H. pylori Growth

Briefly, 100 µL of the brain heart infusion broth (Becton Dickinson, Cockeysville, MD, USA) containing 0.1% glucose and 10% fetal bovine serum (GE Healthcare, Piscataway, NJ, USA) and 50 µL of Gal-2 solution were added to a 96-well plate. Next, 50 µL of the bacterial suspension diluted to 0.1 of OD_600_ was applied to the plate and was cultured under microaerophilic conditions at 37 °C for 2 days. OD_600_ of the cultures were measured after resuspension by vortexing at 1000 rpm for 5 min. The growth rate of the bacteria was determined by comparing the bacterial OD_600_ before and after incubation.

### 4.6. Bactericidal Effect of Gal-2

Briefly, 100 µL of the bacteria suspension was added to an equivalent Gal-2 solution and incubated at 37 °C for 1 h. CFDA (Dojindo, Kumamoto, Japan) and PI (Dojindo) were added to the mixture at final concentrations of 0.15 mg/mL and 1 µg/mL, respectively, according to the manufacturer’s instructions. The mixture was observed under a microscope (BX51, Olympus, Filter; NIBA and WIG) equipped with a charge-coupled-device camera (Hamamatsu, Shizuoka, Japan).

### 4.7. Preparation of Mouse Gastric Tissue Section

C57BL/6J male mice were purchased from Clea Japan (Tokyo, Japan). The animal experiments were reviewed and approved by the Institutional Animal Care and Use Committee of the Josai University and performed in accordance with the institutional guidelines (JU19091).

The mice were killed by cervical dislocation and the stomach were isolated. After the gastric contents were removed, the stomach was incised along the greater curvature. The specimen was fixed using cold Carnoy’s solution (Ethanol: Chloroform: Acetic acid = 6: 3: 1) for 2 h according to Ota et al. [42]. The fixed tissue was incubated with 10, 15, and 20% sucrose for 2 h, sequentially. The tissue was embedded in O.C.T. compound (Sakura FineTechnical, Tokyo, Japan) and frozen with dry ice/acetone. Tissue sections of 10 µm thickness were obtained using the Cryostat (CM3050S, Leica, Germany) and mounted on MAS coated microscope slides (Matsunami Glass, Osaka, Japan).

### 4.8. PAS Staining and IHC

Gastric tissue specimens were refixed before staining. PAS staining was performed using a staining kit (Mutoh Chemistry, Tokyo, Japan) according to the standard protocol.

For IHC, tissue sections were immersed in 0.3% hydrogen peroxide in methanol at room temperature for 30 min to inactivate endogenous peroxidase. The nonspecific reaction was blocked by incubation with normal goat serum (Vectastain ABC kit, Vector Laboratories, Burlingame, CA) at room temperature for 20 min. The sections were incubated with rabbit antibodies against mouse Gal-2 (1:700; Cloud-Clone Corp. Houston, TX, USA) as a primary antibody at 4 °C overnight and followed with the secondary antibody (Vectastain ABC kit) at room temperature for 1 h. Rabbit IgG against keyhole-limpet hemocyanin (Abcam, Cambridge, USA) was used as an isotype control to exclude nonspecific reactions. To enhance the color reaction, the VECTASTAIN ABC reagent (Vectastain ABC kit) was applied to the sections and incubated at room temperature for 30 min. The specific reactions were visualized using 0.02% 3,3′-diaminobenzidine in 0.05 M Tris-HCl (pH 7.4), containing 0.005% hydrogen peroxide and counterstained with hematoxylin. The visualized reactions were observed using a microscope (BZ-X800, Keyence, Osaka, Japan, 40x/numerical aperture; 0.95).

## Figures and Tables

**Figure 1 ijms-21-02697-f001:**
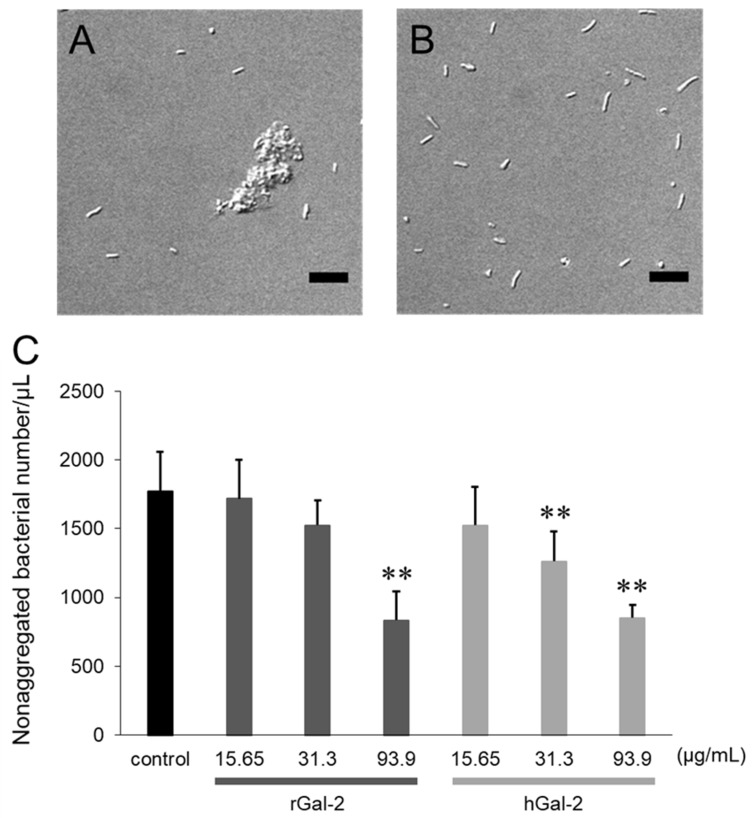
Aggregation of *Helicobacter pylori*-induced by Gal-2. (**A**) *H. pylori* suspension was observed under an optical microscope 1-h after mixing with the rat Gal-2 (rGal-2) solution. (**B**) The bacterial suspension after mixing with phosphate-buffered saline. (**C**) Relationship between bacterial aggregation and concentration of Gal-2. The black, dark gray, and light gray bars represent the control (without Gal-2), rGal-2, and hGal-2, respectively. Scale bar represents 10 µm. Each bar represents the mean ± standard deviation (SD) from five image samples. **, *p* < 0.01 by Dunnet’s test (vs. control).

**Figure 2 ijms-21-02697-f002:**
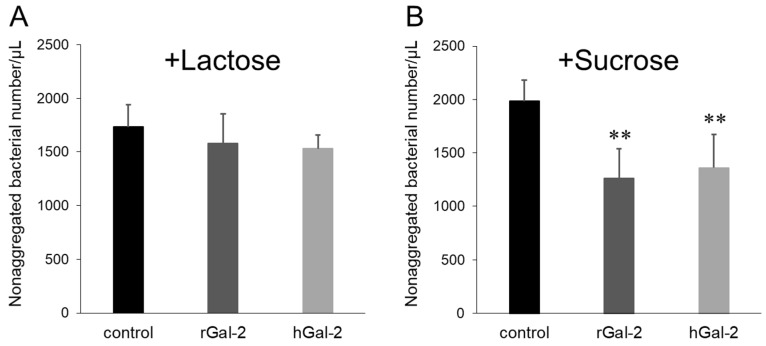
Inhibition of Gal-2 induced *H. pylori* aggregation by competitive sugar, lactose. (**A**) Number of nonaggregated bacteria under the lactose coexistence condition. (**B**) Number of nonaggregated bacteria under sucrose coexistence condition. Concentration of Gal-2 added was 93.9 µg/mL; concentrations of lactose and sucrose were 0.1 M. Black, dark gray, and light gray bars represent the control (without Gal-2), rGal-2, and hGal-2, respectively. Each bar represents mean ± SD from five image samples. **, *p* < 0.01 by Dunnet’s test (vs. control).

**Figure 3 ijms-21-02697-f003:**
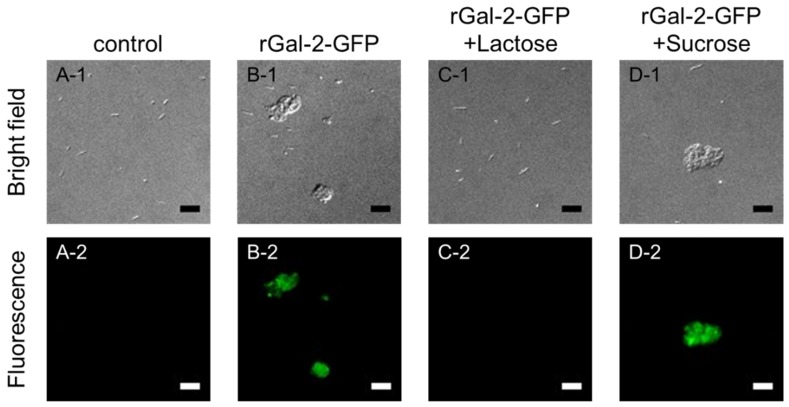
Observation of rGal-2-GFP-*H. pylori* interaction. (**A**) Control (without Gal-2 and sugars). (**B**) rGal-2-GFP alone. (**C**) rGal-2-GFP with lactose. (**D**) rGal-2-GFP with sucrose. Upper (1) and lower (2) images are transmission and fluorescence images, respectively. The concentration of rGal-2-GFP added was 93.9 µg/mL; concentrations of lactose and sucrose were 0.1 M. Scale bars represent 10 µm.

**Figure 4 ijms-21-02697-f004:**
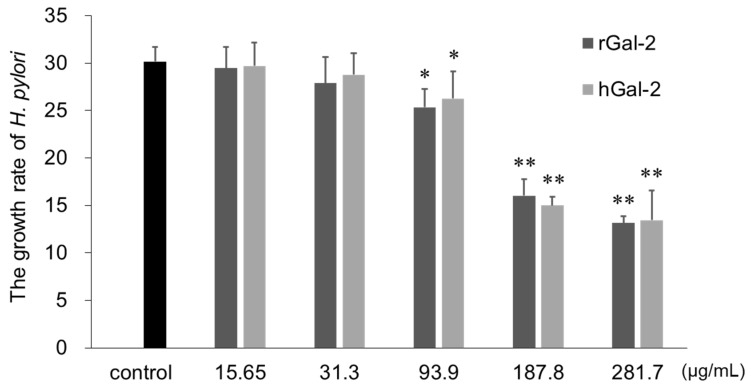
Inhibitory effect of Gal-2 on *H. pylori* growth. The growth rate of *H. pylori* is represented as the optical density ratio at 600 nm (OD_600_) before and after incubation for 3 days. The black, dark gray, and light gray bars represent control (without Gal-2), rGal-2, and hGal-2, respectively. Mean ± SD of the results obtained from four samples. *, *p* < 0.05; **, *p* < 0.01 by Dunnet’s test (vs. control).

**Figure 5 ijms-21-02697-f005:**
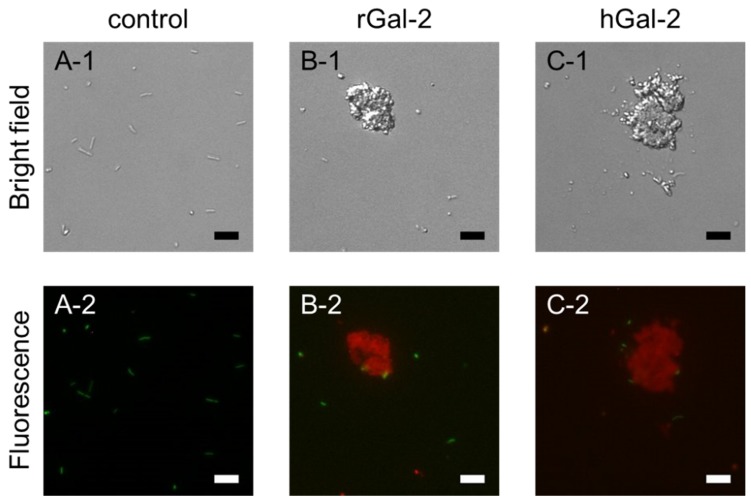
Bactericidal effect of Gal-2 against *H. pylori*. (**A**) Control (without Gal-2). (**B**) rGal-2. (**C**) hGal-2. Upper (1) and lower (2) images are transmission and fluorescence images, respectively. Live and dead *H. pylori* cells were stained with carboxyfluorescein diacetate (CFDA; green) and propidium iodide (PI; red), respectively, for 10 min. The concentration of Gal-2 was 93.9 µg/mL. Scale bars represent 10 µm.

**Figure 6 ijms-21-02697-f006:**
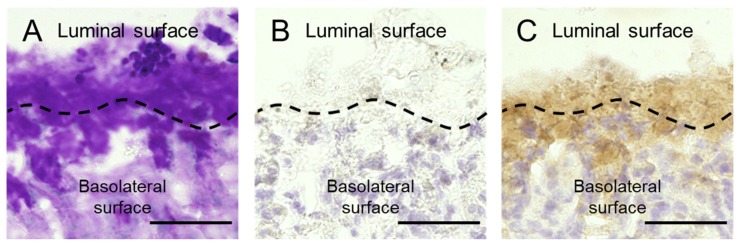
Distribution of Gal-2 in gastric mucus. (**A**) Periodic acid-Schiff (PAS) staining. (**B**) Isotype control. (**C**) Anti-mouse Gal-2 antibody. Upper and lower parts of these images are the luminal side and basolateral surface, respectively. The black dotted lines indicate gastric mucosal surface. Scale bars represent 50 µm.

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
