# Peer review of "Galectin-2 Has Bactericidal Effects against Helicobacter pylori in a β-galactoside-Dependent Manner"

_ijms, 2020, doi:10.3390/ijms21082697_

Round 1

Reviewer 1 Report

The manuscript by Sasaki et al. described that galectin-2 (both rat and human galectin-2) bound to helicobacter pylori through its β-galactoside binding activities and exerted a bacterial killing effect when binding to H. pylori. This is an interesting paper which identified that  galectin-2 is presented in the gastric mucus and potentially exerted bactericidal effects. However, it is way too premature to be published.

  1. The author proved that the purified galectin-2 may interact with the HP to form aggregation through the galactose binding activities of galectin-2. However, there are a lot of carbohydrate-containing substances in the mucus for which some of the galectin-2 may interact. Will the HP aggregate by galectin-2 in the presence of mucus?
  2.  What is the mechanism for galectin-2 kill HP? There were no experiments on why the HP died because of galectin-2 binding. It is possible that is is the secondary effect because of the aggregation which makes the HP hard to obtain nutrition and finally death. A detail molecular mechanism is required.
  3. Will the galectin-2 resistant to the acid environment in the stomach?
  4. In Figure 6, the IHC results can only confirm that galectin-2 expressed in the cells closed to the liminal surface but not galectin-2 in the mucus.

Author Response

Thank you very much for your useful comments on our manuscripts. Our reply to your report is as follows:

Comment #1: The author proved that the purified galectin-2 may interact with the HP to form aggregation through the galactose binding activities of galectin-2. However, there are a lot of carbohydrate-containing substances in the mucus for which some of the galectin-2 may interact. Will the HP aggregate by galectin-2 in the presence of mucus?

reply: We can understand your concern about the interaction of galectin-2 and other carbohydrate-containing substances in the mucus. The present study aims to point out first that galectin-2 shows bactericidal effect against H. pylori by induction of aggregation. We have a next plan to study the effect under in vivo situations. Galectin-3 shows aggregation effect against the bacteria on mucosal surface (Park A. M. et al., Infect. Immun., 84 (4), 1184-1193, 2016). So, we expect that galection-2 have the same effect. Our corresponding statement was included in the text (L. 227-231).

Comment #2: What is the mechanism for galectin-2 kill HP? There were no experiments on why the HP died because of galectin-2 binding. It is possible that is is the secondary effect because of the aggregation which makes the HP hard to obtain nutrition and finally death. A detail molecular mechanism is required.

reply: As you pointed out, the molecular mechanism of bactericidal effect is one of most interesting issues and we also plan some experiments such as metabolome analysis. However, mechanism study is a vast research theme, so that we want to report them in our next article. Corresponding comment was added to L. 213-215.

Comment #3: Will the galectin-2 resistant to the acid environment in the stomach?

reply: Comment about the aggregation effect of galectin-2 in weak acid conditions was added in the text (L. 225-227).

Comment #4: In Figure 6, the IHC results can only confirm that galectin-2 expressed in the cells closed to the liminal surface but not galectin-2 in the mucus.

reply: Because the gastric mucosal surface was unclear, Figure 6 was redrawn.

Reviewer 2 Report

The paper submitted by Takaharu Sasaki et al. describes an interesting role of Gal-2 against Helicobacter pylori. The paper is well written, the work is very interesting, the experiments are particularly well described and the conclusions are correct. No major changes are required. Thus, the paper can be accepted for publication.

Author Response

Thank you very much for your polite review and kind comments on our manuscript.

Reviewer 3 Report

Sasaki et al in the current manuscript show the bactericidal effects of Galectin-2 against Helicobacter pylori. In particular, they have explored the effects of Gal-2 on H. pylori morphology and survival and they showed that the above effects are β-galactoside-dependent.

The study is well written while the experimental approach is straightforward and well designed.

I have two issues that I would like to be addressed in a revised version.

In the methods section please include the numerical aperture of the microscope objectives that have been used.

In the discussion it would be useful to extend your literature and include references like those below:

https://www.ncbi.nlm.nih.gov/pmc/articles/PMC4401781/

https://www.ncbi.nlm.nih.gov/pubmed/21831908/

and comment on the concentration of Gal-2 that have been used. Gal-2 acts as an inhibitor of arteriogenesis both in vivo and in vitro. These findings should be included in cases when the potential use of Gal-2 as a novel antibiotic is discussed.   

Author Response

Thank you very much for your useful comments on our manuscripts. Our reply to your report is as follows:

Comment #1: In the methods section please include the numerical aperture of the microscope objectives that have been used.

reply: According to your comment, the numerical aperture of microscope objectives was added to L. 273-274 and L. 323 in the method section.

Comment #2: In the discussion it would be useful to extend your literature and include references like those below:

https://www.ncbi.nlm.nih.gov/pmc/articles/PMC4401781/

https://www.ncbi.nlm.nih.gov/pubmed/21831908/

and comment on the concentration of Gal-2 that have been used. Gal-2 acts as an inhibitor of arteriogenesis both in vivo and in vitro. These findings should be included in cases when the potential use of Gal-2 as a novel antibiotic is discussed.   

reply: Our comment about the use of galectin-2 as a substitute to conventional antibiotic was added to L. 235-238.

Round 2

Reviewer 1 Report

It is not acceptable for the authors to respond that the molecular mechanisms will be provided in the next manuscript. 

Whether Gal-3 induced clumping of H. pylori is not relevant to Gal-2. The author should provide in Vivo results to confirm the interaction between Gal-2 and H. pylori. The carbohydrate-containing substances are more susceptible to interact with Gal-2 than H. pylori since the H. pylori are minor compared to the mucus. It is not appropriate to just say "further study is required ....".

In lines 235-238, I feel that it is way too much to conclude "Gal-2 can be used as a substitute to conventional antibiotics."